# Role of Innate Immunity in Allergic Contact Dermatitis: An Update

**DOI:** 10.3390/ijms241612975

**Published:** 2023-08-19

**Authors:** Hiroki L. Yamaguchi, Yuji Yamaguchi, Elena Peeva

**Affiliations:** 1Inflammation & Immunology Research Unit, Pfizer, Cambridge, MA 02139, USA; 2Inflammation & Immunology Research Unit, Pfizer, Collegeville, PA 19426, USA

**Keywords:** inducible skin-associated lymphoid tissue, contact hypersensitivity mice, atopic dermatitis, hapten, inflammatory dendritic epidermal cells, antigen presenting cells, ontology, Tc1, Tc2, Tc17

## Abstract

Our understanding of allergic contact dermatitis mechanisms has progressed over the past decade. Innate immune cells that are involved in the pathogenesis of allergic contact dermatitis include Langerhans cells, dermal dendritic cells, macrophages, mast cells, innate lymphoid cells (ILCs), neutrophils, eosinophils, and basophils. ILCs can be subcategorized as group 1 (natural killer cells; ILC1) in association with Th1, group 2 (ILC2) in association with Th2, and group 3 (lymphoid tissue-inducer cells; ILC3) in association with Th17. Pattern recognition receptors (PRRs) including toll-like receptors (TLRs) and nucleotide-binding oligomerization domain (NOD)-like receptors (NLRs) in innate immune cells recognize damage-associated molecular patterns (DAMPs) and cascade the signal to produce several cytokines and chemokines including tumor necrosis factor (TNF)-α, interferon (IFN)-α, IFN-γ, interleukin (IL)-1β, IL-4, IL-6, IL-12, IL-13, IL-17, IL-18, and IL-23. Here we discuss the recent findings showing the roles of the innate immune system in allergic contact dermatitis during the sensitization and elicitation phases.

## 1. Introduction

Contact dermatitis (CD) is one of the most common inflammatory skin diseases caused by exposure to exogenous substances [1]. Prevalence of CD may range from 1.7% to 6.3% with a higher incidence in summer (high temperature and UV), the elderly (skin aging with thinning and skin barrier dysfunction), and females (use of fragrance and metal) [2]. The major risk factors for CD are exposome (non-genetic; environmental) factors that include exposure to irritants and haptens possibly caused by occupational, social, and household activities and skin maceration [3]. Metal exposures can occur not only via direct skin contact (jewelry, mobile phones, and clothes) but also internal intake (metal-rich food, dental implants, orthopedic implants, and stents) [4]. Additionally, there may be genetic risk factors for CD that are associated with skin barrier function (decreased filaggrin expression and claudin [5]) and detoxification enzymes (arylamine N-acetyltransferases 1 and 2 (rapid acetylators dominant in CD patients) and glutathione-S-transferases M1 and T1 (combined deletion in patients allergic to organic mercury compounds) [6]).

The main forms of CD are irritant contact dermatitis (ICD), allergic contact dermatitis (ACD) or a combination of the two. Other subtypes of CD include protein CD, photo-allergic CD, and photo-irritant/-toxic CD [1]. Irritants are substances (chemicals and even water) that cause ICD resulting in inflammation of the skin by direct damage to cells of the epidermis and papillary dermis. Irritants are not specific to the immune system and do not require prior sensitization. On the other hand, haptens are low molecular weight chemicals that penetrate the epidermis and bind to proteins in the skin, stimulate the immune system, and induce sensitization as allergens (hapten–protein complexes; haptenated proteins) in ACD. Of note, haptens require prior sensitization. The common and prominent clinical manifestation of ACD is pruritus, whereas the main symptoms of ICD are a burning sensation and pain [7]. Morphologically, both ICD and ACD lesions show erythema, edema, vesiculation (acute stage; usually observed in positive patch tests in ACD), crust, scales (subacute stage), and lichenification (chronic stage). Blisters and erosions are sometimes observed in ICD lesions. ACD lesions sometimes extend beyond the contact site, whereas ICD lesions are limited to the contact site [7].

ICD initiates when irritants disrupt the skin barrier function and induce spongiosis (intercellular edema) and even necrosis, resulting in the production of damage-associated molecular patterns (DAMPs). Innate immune cells then recognize the DAMPs via pattern recognition receptors (PRRs) including toll-like receptors (TLRs) and produce various cytokines, chemokines, and neuropeptides including tumor necrosis factor (TNF)-α, interferon (IFN)-α, interleukin (IL)-1β, IL-6, IL-12, IL-18, IL-23, and substance P [1]. Damaged keratinocytes also directly produce these innate immune cytokines (IL-1α, IL-1β, IL-6, IL-8, and TNF-α) and cause additional skin tissue damage [8].

Innate immune cells including Langerhans cells, dermal dendritic cells, macrophages, mast cells, innate lymphoid cells (group 1 (natural killer cells; ILC1), group 2 (ILC2), and group 3 (lymphoid tissue-inducer cells; ILC3)), neutrophils, eosinophils, and basophils have emerged as important cell types that contribute to ACD development [9,10,11]. Here we discuss the updated pathogenesis of ACD and the recent findings focusing on the roles of the innate immune system in ACD during the sensitization and elicitation phases.

## 2. Innate Immune Cells in Allergic Contact Dermatitis (ACD)

### 2.1. Pathogenesis of Allergic Contact Dermatitis (ACD)

ACD pathogenesis consists of the sensitization phase (in which the immune system is primed to the allergen) and the elicitation phase (the subsequent immunological response to the re-exposed allergen), and is known as type IV (delayed-type) hypersensitivity reaction [1,7,8,12,13,14].

#### 2.1.1. Sensitization Phase

The sensitization phase is known as an afferent or induction phase, where a hapten makes contact with the skin locally (or systemically via the gut or airway). Haptens are chemicals that are commonly available as patch test agents including, but not limited to, nickel (nickel sulfate), cobalt (cobalt chloride), gold (gold sodium thiosulfate), fragrance (fragrance mix I, fragrance mix II, limonene, linalool, lyral, peppermint, jasmine), and latex (black rubber mix) [15]. Urushiol is also a hapten (plant-derived lipid; non-protein allergen) that is observed in poison ivy, oak, or sumac and is listed in the Japanese Baseline Series with a 9.1% positivity rate for 0.002% concentration (N = 5865) [16]. Urushiol is not listed in the American Contact Dermatitis Society Core Allergen Series [15], but there is ongoing activity to develop a patch test in the US [17].

A hapten penetrates the stratum corneum, conjugates with a self-protein or amino acids (a process called haptenization) via covalent modification (nonmetal haptens) or reversible complex binding (metal haptens), and then forms a hapten-protein complex (or haptenated protein) to serve as an allergen that stimulates the immune system and induces sensitization. Prehaptens and prohaptens are haptens that are activated exogenously (via air oxidation or UV) and endogenously (via enzymes including cytochrome P450), respectively. The degree and severity of ACD may be determined by (1) hapten types (metals; fragrances), sizes, lipophilic/electrophilic nature, and concentrations, (2) exposed skin conditions (barrier function; thickness; maceration; basic skin diseases/comorbidities including ICD, atopic dermatitis, and stasis dermatitis), (3) exposed body sites (face; dorsal hands; mucous), (4) environmental factors (exposome) including occupation, psychological status, lifestyle, nutrition, medication, pollutants, and climate, (5) regulatory T cell function, and (6) genetic factors including detoxification enzymes and ethnicity [1].

Antigen-presenting Langerhans cells (LCs) and dermal dendritic cells (dDCs) mediate innate immunity and initiate adaptive immune reactions with T cells [14]. LCs (and dDCs) monitor the environment with extended dendrites and engulf haptens and/or hapten-protein complexes. As briefly described in ICD pathogenesis, LCs (and dDCs) recognize DAMPs via PRRs, secrete various cytokines, chemokines, and neuropeptides, and stimulate surrounding keratinocytes to secrete these factors as well [1]. Recent findings regarding the roles of LCs and dDCs in the sensitization phase of ACD will be discussed later in Section 2.2.1 and Section 2.2.2; under the influence of these factors (especially IL-1β and TNF-α [18]), LCs and dDCs migrate from the skin to the draining lymph node and present the major histocompatibility complex class 1 (MHC-I) and class 2 (MHC-II) that activate naïve CD8+ and CD4+ T cells (priming), respectively. This priming induces naïve T cells to express hapten-specific T cell receptors (TCRs) that leads to a clonal expansion of hapten-specific T cells, followed by the formation and distribution of memory and effector T cells throughout the body including the skin, lymph nodes, and blood/lymphatic vessels.

#### 2.1.2. Elicitation Phase

The elicitation phase is known as an efferent or challenge phase, where the hapten interfaces with the same region of skin after the sensitization phase is completed and leads to the clinical appearance of acute ACD (erythema; papulation; vesiculation) possibly within 24~72 h [13]. Cytotoxic CD8+ T cells, considered as the primary and key hapten-specific effector T cells, are localized both in the epidermis and dermis and are responsible for histological changes (epidermal-dermal interface dermatitis with spongiosis and inflammatory cell infiltration mainly consisting of lymphocytes and neutrophils) [19]. Other hapten-specific effector CD4+ T cells (including helper T [Th] cells and regulatory T cells) also play roles in (1) the initiation of migration of inflammatory cells from the blood to the local tissue, (2) the induction of apoptosis in keratinocytes, and (3) the reactivation/activation and regulation of inflammatory cells [1]. Th1 cells are the most well-studied effector CD4+ T cells [20], but there is increasing evidence describing the role of Th17 [1] and Th22 cells in the elicitation phase. Additionally, Guttman-Yassky’s group advanced our understanding of ACD pathogenesis by investigating 15 common haptens with molecular and cellular analyses in ACD patients (N = 24) [21]. Skin biopsies from positive patch test results revealed that different haptens exhibit unique immune signatures. Nickel demonstrates the highest immune activation with the induction of pathways associated with innate immune cells, Th1, Th17, and Th22, whereas fragrance and latex, to a lesser extent, induce a potent Th2 and some Th22 polarization with a weaker Th1 and Th17 pathway expression, suggesting that the pathogenesis of ACD is heterogeneous depending on the specific haptens. Her lab also reported similar findings from skin biopsies for four common sensitizers (nickel, dust mite, diphencyprone, and purified protein derivative (N = 10 each)) [22]. Immune signatures are unique in that (1) diphencyprone induces the strongest immune responses, as measured by innate immunity (IL-1α, IL-8), Th1 (IFN-γ, CXCL10), Th2 (IL-5, CCL11), Th17 (LL37), and regulatory T cells (FoxP3, IL-34, IL-37), and (2) dust mites induce modulation of the Th2 pathway, suggesting the heterogeneity of ACD pathogenesis.

Patients with atopic dermatitis show a higher incidence of ICD and ACD because of the (1) increased permeation of irritants and haptens/allergens due to skin barrier dysfunction, (2) active innate immunity including increased access of haptens/antigens to LCs and dDCs, and (3) selective upregulation of the Th2, Th17 (acute), and Th22 (chronic) axis [8]. The association between atopic dermatitis and ACD may be complex with conflicting results [12], but recent studies favor the higher incidence of ACD in atopic dermatitis patients [23], despite a trend toward weaker clinical and immunological ACD reactions [24]. In addition to the heterogenous nature of ACD pathogenesis, atopic dermatitis is heterogenous based on age and ethnicity [25]. Based on flow cytometry analyses in blood, 0–3 year-old patients with atopic dermatitis (N = 29) show suppressed and delayed skin homing Th1 development, as compared with healthy 0–3 year-old controls (N = 14), 3–6 year-old patients with atopic dermatitis (N = 13), and adult patients with atopic dermatitis [26]. Based on molecular profiling analyses of atopic dermatitis lesional and non-lesional skin, African Americans (N = 15) show reduced Th1 and Th17 signatures, as compared with European Americans (N = 15) [27]. Taken together, testing the following hypotheses may be informative: Th2-biased ACD (like fragrance-induced ACD) may be more prevalent and more severe in patients with atopic dermatitis than patients with non-atopic dermatitis, whereas Th1-biased ACD (like nickel-induced ACD) may be less prevalent and milder in patients with atopic dermatitis. Additionally, Th2-biased ACD may be more prevalent and more severe in African American and pediatric patients with atopic dermatitis than European American and adult patients with atopic dermatitis since Th1 bias is reduced and Th2 bias is expected to be increased in African American and pediatric (especially infant and toddler) populations, as compared with European American and adult populations. When ACD pathogenesis is investigated, it may be intriguing to examine the Th1, Th2, and Th17 statuses depending on specific haptens, atopic dermatitis comorbidity, and demographic factors including sex, age, and ethnicity [23].

Recent findings show innate immune cells are actively involved in ACD development not only in the sensitization phase but also in the elicitation phase. Whereas TLRs are transmembrane receptors, nucleotide-binding oligomerization domain (NOD)-like receptors (NLRs) are a group of PRRs located in cytoplasm as “intracellular” proteins that recognize DAMPs inside of innate immune cells and keratinocytes. The NLRP3 (NLR family pyrin domain-containing 3) inflammasome consists of NLRP3 (as a sensor molecule), the adaptor protein, and the cysteine protease pro-caspase 1. The key role of the NLRP3inflammasome in ACD is the production of IL-1β and IL-18 [28]. More specifically, the nuclear factor-κB (NFκB) is activated via TLRs in response to DAMPs, followed by the direct activation of NLRP3 and the production of pro-IL-1β and pro-IL-18. Then, indirect activation of NLRP3 occurs in response to multiple DAMPs including ROS (reactive oxygen species), low molecular weight hyaluronic acid, ATP (adenosine triphosphate), mitochondrial DNA, cardiolipin, uric acid, and cathepsins, followed by the adaptor protein and pro-caspase 1 recruitment. Subsequently, when NLRP3 is fully activated, pro-caspase 1 transforms into caspase 1 that converts pro-cytokines into functional IL-1β and IL-18 [28].

Unfolded protein response (UPR) is a response associated with endoplasmic reticulum stress that maintains the homeostasis of the proteomes by regulating protein synthesis, translocation, post-translational modifications, folding, and the degradation of less toxic protein aggregates at steady state. This UPR possesses three sensors: inositol requiring enzyme-1 (IRE-1), PKR like ER kinase (PERK), and activating transcription factor 6 (ATF6); all of which are inactive when the chaperone protein (GRP-78) is bound to the sensors but are activated when the chaperone detects misfolded proteins. This UPR process is involved in inflammatory diseases including ACD [29]. Haptens, allergens, and sensitizers activate these three sensors. Gendrisch et al. reported that (1) various sensitizers including oxazolone activate IRE-1 and PERK signaling in murine and human keratinocytes and (2) UPR inhibitors targeting IRE-1 and PERK block the translocation and activation of NFκB, the production of IL-6 by keratinocytes, and the CD86 expression level in keratinocyte/DC (human monocytic DC-like cell line THP-1) co-cultures. They also reported that topical application of UPR inhibitors reduces contact hypersensitivity in a mouse model [30].

Nuclear factor erythroid-2-related factor 2 (Nrf2; basic leucine zipper transcription factor) is a master cytoprotective transcription factor that regulates oxidative stress (and is possibly involved in the detoxification and excretion of both organic xenobiotics and metals) [31,32]. Kelch-like ECH-associated protein 1 (Keap1) binds to Nrf2 and migrates into the nucleus from the cytoplasm in response to various stimuli including oxidative stress. Nrf2 regulates neutrophil recruitment and accumulation in the sensitization phase. Additionally, Nrf2-deficient mice are susceptible to ICD and the intensity of ACD is correlated with that of ICD, suggesting that Nrf2 possesses anti-inflammatory effects in ACD [31,32].

Finally, an ACD reaction is resolved after the removal of haptens/allergens/antigens and the recruitment and activation of FoxP3+ regulatory T cells [33]. The Nrf2–Keap1 pathway in innate immune cells may be involved in this post-elicitation phase, known as the resolution or regulation phase [13].

### 2.2. Roles of Innate Immune Cells in Allergic Contact Dermatitis (ACD)

As described in Section 2.1, innate immunity plays a key role in ACD development not only in the sensitization phase mainly via LCs and dDCs but also in the elicitation phase by controlling inflammation via the (1) NLRP3–inflammasome pathway [28], (2) NFκB (nuclear factor kappa-light-chain-enhancer of activated B cells) pathway, (3) UPR pathway [29], and (4) Nrf2–Keap1 pathway (suppression of inflammation) [31,32].

Innate immune cells that are involved in ACD pathogenesis include LCs and dDCs mainly in the sensitization phase and macrophages, mast cells, innate lymphoid cells, neutrophils, eosinophils, and basophils mainly in the elicitation phase (Figure 1). Innate immune cells have emerged as important cell types that contribute to ACD pathogenesis [9,10,11]. Recent findings for each cell type will be discussed.

#### 2.2.1. Roles of Langerhans Cells (LCs) in ACD

Since Paul Langerhans’ discovery of LCs in the human epidermis in 1868, LCs are characterized as epidermal dendritic cells that express Langerin (CD207; C-type lectin receptor that is required for hapten/antigen recognition), CD1a (that presents lipids to CD1a-restricted T cells and is structurally associated with MHC-I), CD1c, EpCAM (epithelial cell adhesion molecule; CD326), E-cadherin, and Birbeck granules (internalized Langerin with hapten/antigen) with negative DC-SIGN (dendritic cell-specific intercellular adhesion molecule-3-grabbing non-integrin; CD209) (Figure 2) [34,35]. There has been a debate regarding the classification of LCs. LCs possess two features: LCs can be functionally dendritic cells since LCs capture hapten, migrate from the epidermis to the draining lymph nodes, and present the hapten/antigen to naïve T cells, followed by activation with increased expression levels of CD28; LCs can be ontogenically/phylogenetically macrophages since LCs share a common precursor (yolk sac; fetal liver) with macrophages during embryogenesis and have auto-renew capabilities [34,35]. Lineage tracing experiments in mice also support that LCs have dual identities by expressing both *Zbtb46* and *Mafb* as a dendritic cell marker and macrophage marker, respectively [36]. To date, no report clearly confirms the dual identities of LCs in human [37]. Instead, inflammatory dendritic epidermal cells (IDECs) have been identified in human inflamed skin (but not in mouse skin at steady state or during inflammation) [38]. IDECs (negative Langerin) reside in the basal keratinocyte layer (Stratum Basale) within the epidermis, whereas LCs reside in the supra-basal keratinocyte layer (Stratum Spinosum) and penetrate the epidermal tight junction barrier to capture antigens (Figure 1). Both possess FcεRI (a high-affinity IgE receptor that drives ACD upon haptens/allergens and is expressed on LCs, IDECs, mast cells and basophils) below the tight junctions (possibly to capture the penetrated haptens/allergens/antigens), thereby serving as antigen-presenting cells [37,38]. IDECs can be “monocyte-derived” precursors of LCs in the same way as macrophages, but further investigations are necessary.

PRRs expressed on LCs (derived upon isolation from normal human skin) include TLR1, TLR2, TLR3, TLR5, TLR6, and TLR10, suggesting the lack of expression of TLR4, TLR7, TLR8, and TLR9 [39]. LCs differentially respond to multiple TLR agonists that mimic DAMPs. TLR2 agonists (peptidoglycan, PGN (also known as TLR6 agonist) and lipoteichoic acid, LTA) trigger LC maturation by upregulating the expression of maturation markers (CD25, CD83, and DC-LAMP) and costimulatory molecules (CD80 and CD86) (Figure 2); whereas the TLR4 agonist (lipopolysaccharide, LPS) has only marginal effects on LC maturation and on the production of IL-10, IL-8, and TNF-α (despite a slight upregulation of IL-6 production) possibly due to lack of TLR4 expression on human LCs [39]. The production of IL-6, IL-8, and TNF-α is most remarkable when LCs are stimulated with the TLR3 agonist (Poly I: C); whereas that of IL-10 is highest when stimulated with TLR6 agonist (PGN stimulates IL-10 production whereas LTA does not) (Figure 2) [39].

TNF-α and IL-1β stimulate LCs to lose their adhesion with the surrounding keratinocytes (by decreasing the expression levels of adhesion molecules including EpCAM and E-cadherin) and enhance their migration into the lymph nodes (Figure 2) [34]. LCs may induce various types of T cells as well as CD4+ Th1 cells (and type 1 CD8+ T [Tc1] cells that are generated in response to IL-2 and IL-12 and secrete IFN-γ) in the draining lymph nodes (Figure 2) [40].

Experiments with human CD1a-transgenic mice show that urushiol triggers CD1a-dependent skin inflammation driven by Th17 cells (secretion of IL-17) and Th22 cells (secretion of IL-22) (Figure 2) [41]. Furthermore, in an in vitro co-culture system, CD1a-transfected antigen-presenting cells (K562 cells) pulsed with urushiol activate the production of IL-17 and IL-22 by CD4+ T cells isolated from blood of patients with ACD caused by poison ivy (N = 6), suggesting an important role of LCs in CD4+ T cell response to urushiol (differentiation into Th17 and Th22 cells) in a CD1a-restricted manner [41]. Another study using a murine infection model also shows that LC-derived IL-6 is required for the induction of Th17 cells via CLEC7A (C-type lectin receptor for fungal recognition; Dectin-1) on LCs (Figure 2), whereas Th1 differentiation takes place in the absence of CLEC7A1 ligation (Figure 2) [42].

By using a Langerin-diphtheria toxin (DT) receptor knock-in mice, human Langerin-DT-A transgenic mice, and mice deficient in TSLPRs (thymic stromal lymphopoietin receptors) in LCs, Nakajima et al. reported that LCs initiate sensitization with protein antigens (ovalbumin) and induce Th2 immune responses via TSLP signaling (Figure 2) [43]. Human LCs are also potent activators of naïve CD4+ T cells to differentiate into Th2 cells (secretion of IL-4, IL-5, and IL-13) [44]. Additionally, human LCs efficiently prime naïve CD8+ T cells possibly via IL-15 that is secreted by LCs and is localized to the immunologic synapse of the LC and T cell, followed by the selection and expansion of antigen-specific cytotoxic CD8+ T cells (Figure 2) [45]. Blocking IL-15 (or the addition of IL-10) during the co-culture of LCs and naïve CD8+ T cells attenuates the growth (~50% reduction) and function of the T cells (reduced secretion of IL-2 and IFN-γ; reduced expression of CD107a, granzymes, perforin, and Bcl-2 (required for keratinocyte apoptosis)) [45].

LCs also have capabilities to induce regulatory T cells and T cell anergy (tolerance mechanism by which T cells are functionally inactivated in response to antigens and remains alive with growth arrest) under various conditions including protein antigen sensitization, UV or ionizing radiation, and self-antigen expression on LCs or keratinocytes (Figure 2) [40]. In this sense, LCs may be involved in the post-elicitation/resolution/regulation phase of ACD. Using DNTB (2,4-dinitrothiocyanobenzene; a weak sensitizer as compared with DNFB (2,4-dinitrofluorobenzene)) as a hapten in mice, Gomez de Aguero et al. reported that LCs induce tolerance with the weak hapten, DNTB, by migrating into the draining lymph node and presenting the hapten to CD8+ T cells [46]. The depletion of LCs breaks tolerance to DNTB by allowing priming of the CD8+ T cells. Their adaptive transfer experiments also showed that (1) DNTB presentation by LCs to CD8+ T cells reduces their capability to produce IFN-γ and (2) preexisting CD4+Foxp3+ regulatory T cells are activated by LCs and are essential for LC-mediated tolerance [46]. Using only autologous human skin, Seneschal et al. reported that (1) LCs selectively induce the activation and proliferation of regulatory T cells in the absence of exogenous antigens and (2) LC-mediated regulatory T cell proliferation may be antigen-specific probably via MHC-II (more specifically HLA-DR and HLA-DP), costimulatory signal CD80 and CD86, IL-2 (associated with T cells), and IL-15Rα (associated with LCs), but not via CD1a nor MHC-I. They also reported that (3) the same LCs activate and induce the proliferation of both regulatory T cells and skin pathogen-specific effector T cells (secreting IFN-γ and IL-17) and finally (4) LC-mediated effector T cell proliferation may be regulated via MHC-II, CD80 and CD86, IL-2, IL-15Rα, and CD1a, but not via MHC-I, suggesting the role of LCs in negatively and positively modifying immune response [47].

Consequently, the roles of LCs in ACD include (1) activation, maturation, and migration into the draining lymph node in response to DAMPs via PRRs, and finally the presentation of haptens/antigens to naïve T cells, (2) induction of cytotoxic CD8+ T cells and effector CD4+ T cells including Th1, Th2, Th17, and Th22 cells, and (3) induction of regulatory T cells and T-cell anergy, thereby inducing tolerance.

#### 2.2.2. Roles of Dermal Dendritic Cells (dDCs) in ACD

DCs are innate immune cells derived from bone marrow (arising from lympho-myeloid hematopoiesis) and activate adaptive immune reactions by T cells in response to DAMPs and pathogen-associated molecular patterns [34,48,49,50]. Cutaneous DCs at steady state can be categorized in multiple ways but may be classified into two types: LCs in the epidermis and dermal DCs (dDCs). The dDCs at steady state are conventional DCs (cDCs), also known as myeloid DCs, that may be subclassified into two types (Figure 3): cDC1s (type 1) and cDC2s (type 2). Plasmacytoid DCs (pDCs) may be classified as dDCs since they are observed in inflammatory skin diseases although they circulate in the blood and are also found in peripheral lymphoid organs. Monocyte development is different from the above mentioned dDCs in that monocytes are not derived from common DC progenitors although monocytes and DCs do share macrophage-DC precursors [35,48]. However, monocyte derived DCs (Mo-DCs) may be included as dDCs, especially when the pathogeneses of inflammatory skin diseases are investigated.

Using single-cell RNA sequencing from the DC-containing population (HLA-DR+ cells) from human blood and cytometry by time-of-flight analyses (CyTOF; combined technique of mass spectrometry and flow cytometry to enable multiple protein (antigen) analyses at single-cell level [51]), See et al. elucidated a continuous differentiation process that emerges in the bone marrow with common DC progenitors: (1) DC precursors (pre-DCs) and pDC share common markers including CD123, CD303, and CD304; (2) pre-DCs (IL-12/p40 production and induction of naïve CD4+ T cell proliferation) are functionally different from pDCs (IFN-α production); (3) pre-DCs differentiate into early pre-DCs, pre-cDC1s, and pre-cDC2s; and (4) pre-cDC1s and pre-cDC2s are functional, as compared with cDC1s and cDC2s, in terms of functional TLR9 expression, TNF-α and IL-12/p40 productions in responses to TLR9 agonists (pre-cDC1s > pre-cDC2s), and expression levels of costimulatory molecules (CD40, CD80, CD83, and CD86) [50].

Based on ontological and functional analyses, various markers are available to identify cDC1s, cDC2s, pDCs, and Mo-DCs [34,48,49,50]. Human cDC1s and cDC2s express interferon regulatory factor 8 (IRF8) and IRF4 as transcription factors, respectively, whereas pDCs express both IRF4 and IRF8 and Mo-DCs express neither IRF4 nor IRF8 at steady state [34]. However, Mo-DCs express IRF4, CD11c, and CD1c during inflammation, indicating the similarities between cDC2s and Mo-DCs [34]. The functional roles of each dDC type are discussed in the next subsections with recent findings.

##### Roles of Conventional Dendritic Cells 1 (cDC1s) in ACD

Overall, cDC1s possess potent capabilities to present antigens, activate CD8+ T cells via MHC-I, and activate Th1 and natural killer cells via IL-12, suggesting key roles in anti-tumor and anti-virus immunity (Figure 3) [34]. In response to haptens and protein antigens, cDC1s help differentiate naïve T cells into Th1 (DNFB; *C. albicans*) and Tc1 (type 1 CD8+ T cells; DNFB; OVA) in ACD mouse models [40]. As PRRs, cDC1s express TLR3 (that recognizes double-strand RNA and induces production of IFN-α via IRF3), TLR9 (that recognizes DNA and induces production of IFN-α), TLR10 [34], TLR11, and TLR12 [52]. Of note, the limited expression of TLR10 in cDC1s may be associated with a negative regulation of T cell activation [53]. TLR2 and TLR4 may be expressed on mature CCR7+ cDC1s that use IDO1 (the tryptophan metabolic enzyme indoleamine 2,3-dioxygenase 1) and possess tolerogenic activity (Figure 3) [54]. The C-type lectin CLEC9A is also a key receptor expressed on cDC1s that recognizes F-actin filaments derived from necrotic cell death [34].

In the oncology research field, Giampazolias et al. reported that secreted gelsolin suppresses the capability of cDC1s to recognize F-actin from necrotic cells via CLEC9A and to cross-present tumor antigens to CD8+ T cells via MHC-I by using gelsolin-deficient mice and by analyzing human cancer tissues (inverse correlation between gelsolin expression and patient survival; positive correlation between low gelsolin expression and patient survival with mutations in F-actin-binding proteins) (Figure 3) [55]. A similar escape mechanism by gelsolin may exist in healthy subjects who do not experience ACD, whereas patients with ACD may show a decreased level of gelsolin.

##### Roles of Conventional Dendritic Cells 2 (cDC2s) in ACD

cDC2s form a heterogenous population with a wide range of markers, but CD1c (also known as BDCA1) may be a key differentiation marker (whereas cDC1s express CD141/BDCA3) (Figure 3) [52]. Overall, cDC2s present haptens/antigens via MHC-II, followed by the priming and activation of helper CD4+ T cells (especially Th2 and Th17), suggesting roles in immunity against haptens/allergens, parasites, and extracellular pathogens. More specifically, cDC2s help differentiate naïve T cells into Th2 (FITC + dibutyl phthalate [DBP]; OVA + papain; house dust mite in an IRF4 dependent manner [56]), Th17, Th1, and Tc1 (FITC + DBP) (Figure 3) [40]. cDC2s possess a wide range of PRRs including TLRs (TLR2, TLR4, TLR5, TLR6, and TLR8) and NLRs (NOD2, NLRP1, NLRP3, and NAIP). cDC2s also express C-type lectins CLEC4A, CLEC6A, CLEC7A (Dectin-1), CLEC10A, and CLEC12A. cDC2s secrete IL-12, IL-23, IL-1, IL-8, IL-10, and TNF-α [40]. 

The key role of cDC2s is the generation of Th2 immunity (Figure 3) [57]. By using a mouse model with triple mutations of NFIL3 (nuclear factor, IL-3 regulated), C/EBPα, and C/EBPβ (transcription factors that interact with the CCAAT [cytosine-cytosine-adenosine-adenosine-thymidine] box motif), all of which are enhancers of ZEB2 (Zinc finger E-box-binding homeobox 2; transcription factor for TGF-β signaling), Liu et al. reported that (1) the triple mutations ablate ZEB2 expression in common DC progenitors, but not lymphoid progenitors, causing the complete loss of pre-cDC2 specification and mature cDC2 development and (2) the cDC2 ablated mice do not generate Th2 immunity in response to helminth infections [57]. Mayer et al. reported that (1) IL-13 signaling differentiates cDC2s in the dermis in a STAT6 dependent manner, whereas cDC2s in lung and small intestine are STAT6-independent and (2) without IL-13 signaling, dermal cDC2s are stable in number and show minimal response to allergens with a diminished ability to support Th2 differentiation, whereas Th17 cells are increased. They also reported that (3) human dermal cDC2s (from healthy volunteers) express an IL-4 and IL-13 gene signature (Figure 3) that is not observed in cDC2s in the blood, spleen, and lung [58]. These results indicate that the generation of Th2 immunity by cDC2s in skin does not require immune networks including IL-33 (that is required in the lung), IL-25, TSLP, or skin microbiota probably because skin ILC2s (group 2 innate lymphoid cells; Section 2.2.6) produce IL-13 at steady state [59]. 

By combining transcriptional (single-cell and bulk RNA-sequencing) and chromatin (an assay for transposase-accessible chromatin [ATAC]-sequencing) analyses with genetic reporter expression analyses, Brown et al. identified two principal cDC2 lineages: cDC2A and cDC2B [52]. cDC2As show T-bet+ (known as the transcription factor observed in Th1 cells), CD86 low, AREG high (amphiregulin; epidermal growth factor family), and MMP9 high (matrix metalloproteinases-9). Human spleen CD1c low CLEC10A- CLEC4A high cDC2s are relevant to mouse cDC2As. On the other hand, cDC2Bs show T-bet-, RORγT+ (known as the transcription factor observed in Th17 cells), and IL1B high. cDC2Bs increase IL-6 and TNF-α in response to CpG (TLR9 agonist) stimulation, express higher levels of TLR6, TLR8, and TLR9 than cDC2As, and induce Th1 and Th17, suggesting a potent pro-inflammatory phenotype. Human peripheral blood and spleen CD1c+ CLEC10A+ CLEC4A low cDC2s are relevant to mouse cDC2Bs [52].

Breed et al. reported that (1) CD301b+ cDC2s exhibit a Th2 cytokine signature and require steady-state IL-4 receptor signaling in the thymus and (2) selective ablation of CD301b+ cDC2s impairs clonal deletion without affecting regulatory T cells using transcriptional and phenotypic analyses, suggesting the role of cDC2s in promoting central tolerance in the thymus via Th2 cytokines (IL-4 and IL-13) and the IL-4 receptor pathway (Figure 3) [60]. This finding may partly explain why ACD severity is attenuated in atopic dermatitis patients. The tolerogenic IDO1 pathway is expressed in mature CCR7+ cDC1s but not in cDC2s (refer to Roles of Conventional Dendritic Cells 1 (cDC1s) in ACD). Gargaro et al. reported that mature CCR7+ cDC1s induce regulatory cDC2s via tryptophan metabolite L-kynurenine using an experimental autoimmune encephalomyelitis model [54], indicating that this pathway may be also available in ACD pathogenesis.

In the oncology research field, a recent review by Saito et al. outlines the role of cDC2s in tumor immunity [61]. Binnewies et al. identified cDC2s that present tumor-derived antigens to CD4+ T cells but fail to support the anti-tumor differentiation of CD4+ T cells in tumor draining lymph nodes of mice [62]. They also reported that (1) regulatory T cell depletion enhances the capability of cDC2s to differentiate CD4+ T cells, ensuring the anti-tumor activities; and (2) intratumor cDC2s density positively correlates with abundant fully functional CD4+ T cells and with responsiveness to anti-PD-1 (programmed cell death protein-1) therapy in treating melanoma patients with low regulatory T cell abundance [62]. By using a mouse lacking cDC1s, Duong et al. reported that (1) ISG+ (IFN-stimulated genes positive) cDC2s activate CD8+ T cells ex vivo. They also reported that (2) IFN-β produced by regressor tumors drive the ISG+ cDC2 state via MHC-I dressing and costimulatory molecule activation (CD40, CD80, and CD86), activate tumor-reactive CD8+ T cells, and rescue anti-tumor immunity against progressor tumors. They finally reported that (3) the ISG+ cDC2s gene signature is detectable in human tumors [63]. These results observed in immune-oncology research suggest that studying the relationship between cDC2s and regulatory T cells and/or between ISG+ cDC2s and CD8+ T cells may be informative to elucidate the pathogenesis of ACD.

##### Roles of Plasmacytoid Dendritic Cells (pDCs) in ACD

pDCs produce IFN-α via TLR7 as PRR and TNF-α and IL-6 via TLR9 in response to single-stranded RNA and double-stranded DNA, respectively, and may be associated with inflammatory/autoimmune skin diseases including psoriasis and lupus [34]. pDCs express MHC-II and costimulatory molecules including CD40, CD80, and CD86, and prime Th1, Th2, and regulatory T cells. However, the capability to present antigens to naïve CD4+ T cells may not be as efficient as cDCs.

Miller et al. indicate that a balance of cDC1s and cDC2s in the draining lymph node may be crucial to maintain homeostasis and avoid Th2 responses (to haptens/antigens including fluorescein isothiocyanate [FITC]) rather than targeting pDCs [64]. Using DT receptor transgenic mice driven by the promoter of human C-type lectin CLEC4C, they reported that (1) DT treatment (the disruption of CLEC4 promoter, causing depletion of TCF4 (master transcription factor for pDCs)-expressing cells) augments Th2-dependent skin inflammation in a contact hypersensitivity model induced by FITC (but no effects on the Th1-biased model using DNFB), (2) the Th2-biased response is independent of reduced IFN-α/IFN-β due to pDC depletion, (3) based on single-cell RNA-sequencing of DCs, DT treatment also depletes Th1-priming cDC1s (EpCAM+ cDC1s and CD103/integrin αE+ cDC1s that migrate from skin to the draining lymph node and induce Th1 responses), and (4) DT treatment increases Th2-priming cDC2s (CD11b+ cDC2s) in the draining lymph nodes [64]. These findings suggest that maintaining a ratio of cDC1s: cDC2s in the draining lymph nodes in the sensitization phase of ACD may be important to prevent or attenuate Th2-biased ACD including fragrances.

##### Roles of Monocyte Derived Dendritic Cells (Mo-DCs) in ACD

Mo-DCs are ontogenically and phylogenetically different from the dDCs. A lineage tracing of monocyte populations does not lead to DC populations using the fate mapping via Ms4a3 as a specific gene that granulocyte-monocyte progenitors express [65]. However, it is difficult to distinguish Mo-DCs from cDC2s in situ in humans since Mo-DCs share surface markers with cDC2s (HLA-DR+, CD11c+, CD1c+, CD172a+, CLEC6A, CLEC7A), respond to DAMPs, function as antigen presenting cells, and activate T cells (possibly at the site of inflammation; not migrating to lymph nodes). Mo-DCs are derived from blood monocytes and migrate into the tissue in the inflammatory state including skin sensitization.

Since human samples are difficult to obtain except for blood and since the method for how to differentiate Mo-DCs is well established from peripheral blood mononuclear cells using M-CSF, IL-4 and TNF-α [66], Mo-DCs are used to conduct in vitro studies. Hoper et al. reported that (1) Mo-DCs respond to LPS and nickel similarly in terms of the TLR4 gene signature and (2) there was a difference in the Nrf2 target gene and cholesterol biosynthesis gene signatures, suggesting that each hapten/allergen has unique effects on immune cells in vitro [67]. Measuring the activation of human Mo-DCs under coculture conditions with primary human keratinocytes (with or without T cells) is useful to screen potential haptens/allergens (when developing cosmetics considering the ethics associated with the use of experimental animals) [68]. To reduce in vivo experiments, which are currently mandatory for topical drug development to rule out a potential risk of ACD in humans, the use of human Mo-DCs in studying ACD would be informative and valuable provided the Mo-DCs are collected and evaluated appropriately [34].

#### 2.2.3. Roles of Macrophages in ACD

Tissue-resident macrophages are directly derived from the yolk sac/fetal liver during embryogenesis and are replenished from blood monocytes (bone marrow progenitor) [69]. Macrophages (CD68+) can be classified into two types: M1 macrophages (classically activated; Th1 skewed; CD14+ CD16- monocyte derived; HLA-DR+ CD80+ CD86+) and M2 macrophages (alternatively activated; Th2 skewed; regulatory function; CD14+ CD16+ monocyte derived; CD163+ CD206+) (Figure 4). M1 macrophages express TLR4 and produce IL-1β, TNF-α, IL-6, and IL-12 via the NFκB pathway in response to IFN-γ. M2 macrophages can be subclassified into four types: M2a (IL-4 and IL-13 induced; CD163+ CD206+ CD369+), M2b (TLR agonists and IL-1 receptor agonists induced; CD86+; IL-10 production; also known as regulatory macrophages), M2c (induced by IL-10; IL-10 and TGF-β production; removal of apoptotic cells), and M2d (tumor-associated macrophages) [69].

Suwanpradid et al. reported that (1) skin biopsies obtained from ACD patients express iNOS+ (inducible nitric oxide synthase; production of nitric oxide; possibly contributing to ACD associated inflammation) CD14+ M1 macrophages and Arg1+ (arginase 1; anti-inflammatory function) M2b macrophages and (2) iNOS contributes to ACD-associated inflammation in mice. They further reported that (3) mice lacking Arg1 in macrophage show exacerbated ear thickness, dermal inflammation, and increased iNOS and IL-6 expression, as compared with normal littermates, and the attenuated ACD-associated responses with iNOS inhibition (Figure 4) [70]. Otobe et al. reported that CX3CR1 (the receptor for CX3CL1; expressed in monocytes/macrophages, T cells, and NK cells) deficient mice show attenuated ACD-associated responses, as measured by ear thickness in response to DNFB, with decreased expression levels of TNF-α, IL-6, and M1 macrophage markers including iNOS and increased expression levels of M2b macrophage markers including Arg1, suggesting the role of skewed polarization towards M2 phenotype in attenuated contact hypersensitivity response (Figure 4) [71].

M1 macrophages contribute to ACD development in response to IFN-γ (possibly Th1 skewed), whereas M2 macrophages may contribute to ACD development in a Th2-skewed manner (possibly M2a) as well as suppressive effects on ACD (via IL-10 and Arg1; possibly M2b). Virgens et al. reported that skin biopsies from patients with ACD (N = 13; at 48 h of patch testing) caused by methylisothiazolinone and methylchloro-isothiazolinone show pronounced M2 macrophages (CD163+; possibly M2a) and Th2 cells (accumulation in perivascular areas), an increase in IL-4 and IL-13 mRNA expression, a mild increase in IFN-γ m RNA expression, and a decrease in RORC mRNA expression (Figure 4) [72], suggesting a contribution of M2a macrophages in Th2-biased ACD pathogenesis. 

Future studies with M2 macrophage classification may further elucidate the roles of macrophages in ACD pathogenesis.

#### 2.2.4. Roles of Inducible Skin-Associated Lymphoid Tissue (iSALT) in ACD

A recent advancement in imaging studies in vivo has led to a discovery of an immunological unit that is called inducible skin-associated lymphoid tissue (iSALT) for efficient antigen presentation in inflammatory skin diseases including ACD [40,73]. The iSALT is transiently observed close to postcapillary venules and consists of perivascular macrophages, dDCs, and T cells (Figure 1). This cluster formation appears within hours in response to hapten application to murine skin. Keratinocytes secrete IL-1α in response to DAMPs and activate macrophages around post capillary venules, followed by the production of CXCL2 and recruitment of dDCs to the cluster. Leukotriene B4, a lipid mediator, contributes to the cluster formation by enhancing dDC recruitment. Both cDC1s (CD103+) and cDC2s (CD11b+) are observed in the cluster.

Suwanpradid et al. also reported that (1) macrophages secrete IL-27 and activate keratinocytes and dDCs to produce IL-15, followed by the enhancement of BCL2 and the survival of T cells and (2) blocking IL-27 in allergic contact hypersensitivity mice leads to decreased IL-15 production, a decrease in BCL2 expression in T cells, and a decrease in CD8+ T cells and clusters, suggesting the key role of IL-27 produced by macrophages in maintaining T cells within the cluster [74]. Liu et al. also reported that (1) CX3CR1 intermediate monocytes form a cluster around hair follicles and induce keratinocyte apoptosis in a multicolor-labeled murine contact hypersensitivity model using oxazolone, (2) T cells and dDCs migrate to the cluster, (3) T cells secrete IFN-γ, followed by monocyte migration in a CCR2-dependent manner and monocyte cluster formation in a CXCR2-dependent manner, and (4) the CXCR2 antagonist significantly reduces monocyte activation and keratinocyte apoptosis [75].

It is not known whether iSALT is observed in human skin, presents antigens to T cells, and supports T cell activation. However, skin biopsies from patients with ACD show dDC (CD11c+)–T cell (CD3+) clusters underneath the vesicle in the dermal–epidermal junction, implying that the activation of T cells takes place in the cluster [76]. By examining skin biopsies obtained from ACD patients under various conditions (specific haptens and concentrations; sites of the body; demographics; comorbidities; exposome), an iSALT cluster may be confirmed in humans in due course.

#### 2.2.5. Roles of Mast Cells in ACD

Mast cells are derived from mast cell progenitors that differentiate from hematopoietic stem cells in the bone marrow and migrate to the local tissue where they mature in response to various cytokines including stem cell factor, TGF-β, nerve growth factor (NGF), IL-3, IL-4, IL-9, and IL-33 (Figure 5) [77]. The cytoplasm of mast cells contains various granules storing chemical mediators (histamine), proteoglycans, and proteases [78]. The receptors expressed on mast cells include FcεRI, TLRs, Mas-related G-protein-coupled receptor X2 (MRGPRX2; mouse ortholog MrgprB2), and cytokine receptors (c-kit for stem cell factor) [78].

Pruritus may be a behavioral extension of type 2 immunity including atopic dermatitis and ACD [79]. Mast cells play a central role in the pruritus mechanisms via not only the canonical IgE–mast cell–histamine–H1R (histamine receptor 1) axis but also intimate crosstalk with sensory neurons via receptors including the IL-31 receptor α, TrkA (tropomyosin receptor kinase A; high affinity to NGF), and protease-activated receptor 2 (PAR2) (Figure 5) [80]. Engeroll et al. reported that (1) free IgE binds to FcεRI (that drives ACD upon haptens/allergens and is expressed on LCs, IDECs, mast cells, and basophils), whereas the IgE–allergen immune complex binds to CD23 (FcεRII; the low-affinity receptor for IgE; constitutively expressed on B cells) on mast cells and (2) the free IgE–FcεRI pathway initiates ACD (inflammatory), whereas IgE–immune complex–CD23 pathway reduces FcεRI binding and enhances CD23-dependent serum clearance (noninflammatory) [81]. However, targeting the histaminergic pathway with antihistamine drugs is not always efficacious in treating pruritus, suggesting the involvement of a non-histaminergic pruritus pathway (Figure 5; NGF–TrkA and tryptase–PAR2) [79]. Meixiong et al. reported that (1) mast cell activation via Mrgprb2 contributes to non-histaminergic pruritus independently of the IgE–FcεRI–histamine axis in mice and (2) Mrgprb2 agonism enhances the secretion of tryptase and excites a different sensory neuron population (Mrgprd+, Mrgpra3+), as compared with the neurons that respond to the IgE–histamine signal (H1R+). They also reported that (3) a Mrgprb2-deficiency decreases pruritus in ACD nonclinical models and (4) the expression levels of PAMP1-20 (proadrenomedullin N-terminal 20 peptide; MRGPRX2 agonist) are elevated in skin lesions obtained from patients with ACD (N=11) (Figure 5) [82]. The role of MRGPRX2 on mast cells in ACD pathogenesis is further reviewed in detail by Roy et al. [83]. Mast cell degranulation via Mrgprb2 may mediate imiquimod-associated dermatitis including ACD murine model [84]. 

Hoppe et al. reported that mast cells initiate vascular changes (vasodilation, permeabilization, and neutrophil influx) by prompt degranulation in response to cell stress or damage via the ATP–P2X7 pathway (causing histamine-triggered edema) and the IL-33–ST2 (IL-33 receptor) pathway (causing efficient induction of neutrophil infiltration) using various murine ACD models (Figure 5) [85]. Mast cells are important target cells for IL-33 that is released in response to DAMPs. Babina et al. reported that (1) prolonged exposure to IL-33 increases the number of mast cells, (2) IL-33 induces swift p38 phosphorylation and stimulates the expression levels of histidine decarboxylase (enhancing histamine synthesis), and (3) IL-33 attenuates degranulation and the expression levels of FcεRI, possibly due to chronic exposure, suggesting an overall stimulating role of IL-33 in ACD pathogenesis with an anti-inflammatory role at the chronic stage (Figure 5) [86]. Leyva-Castillo et al. reported that IL-13 derived from mast cells selectively suppresses the capability of dDCs to polarize naïve CD4+ T cells into Th1 cells in response to antigen exposure, suggesting the important role of mast cells in Th2-biased ACD development (Figure 5) [87].

Recent studies suggest the immunosuppressive functions of mast cells. Hirano et al. reported that (1) PD-L1 (programmed death ligand 1)-deficient mice show worsened ear swelling with an increased number of IFN-γ+ CD8+ T cells in the skin, (2) the PD-1/PD-L1 pathway is involved in the elicitation phase of ACD in mice, and (3) PD-L1 expression is highest on mast cells based on bone marrow chimera experiments and flow cytometric analysis. They also reported that (4) the administration of an anti-PD-L1 antibody during the elicitation phase significantly worsens ear swelling in wild-type mice, whereas mast cell-deficient mice do not show significant effects in response to PD-L1 depletion, and (5) human skin-derived mast cells show high expression levels of PD-L1 based on the public database (provided by Functional Annotation of the Mammalian Genome 5 project) and immunohistochemical analysis, suggesting the role of PD-L1+ mast cells on negative regulation of CD8+ T cells (Figure 5) [88]. Overall, mast cells may play inhibitory roles in murine ACD models possibly via the maintenance of IL-10-producing regulatory B cells in the skin [89], although there are several controversial findings regarding IL-10 derived from mast cells [90]. The platelet-activating factor (PAF), a soluble lipid, mediates both local acute inflammation (mast cell migration to lymph node; macrophage, neutrophil, and eosinophil chemoattraction; neutrophil extracellular trap release) and delayed systemic immunosuppression (via a PAF receptor on mast cells in a cyclooxygenase-2-dependent and/or a histidine decarboxylase-dependent manner) [91].

Consequently, mast cells exhibit pro-inflammatory (possibly with mild severity; Th2-biased ACD) and anti-inflammatory roles (possibly under severe or persistent conditions; Th1-biased ACD) in the pathogenesis of ACD and ACD-associated pruritus [92].

#### 2.2.6. Roles of Innate Lymphoid Cells (ILCs) in ACD

ILCs have lymphoid morphology and effector functions similar to T cells but lack specific antigen receptors [93,94,95]. Since ILCs and effector T cells share various transcription factors, ILCs can be considered as “innate” counterparts of effector T cells [93,94]. Based on their functions and key transcription factors, ILCs can be classified into three groups (Figure 6): group 1 ILCs (NK (natural killer) cells and ILC1s) in association with Tc1 and Th1 cells, group 2 ILCs (ILC2s) in association with Tc2 (CD8+ cytotoxic T cells that produce type 2 cytokines) and Th2, and group 3 ILCs (ILC3s and LTi (lymphoid tissue inducer) cells) in association with Tc17 (CD8+ cytotoxic T cells that produce type 17 cytokines) and Th17 [93].

ILCs are derived from common innate lymphoid progenitors that originate from common lymphoid progenitors (shared with T cells) in the bone marrow and differentiate into three progenitors: (1) ILC progenitors that differentiate further into ILC1s with T-bet expression (producing IFN-γ and TNF-α), ILC2s with GATA3 expression (producing IL-4, IL-5, IL-9, and IL-13), and ILC3s with RORγt expression (producing IL-17 and IL-22), (2) NK progenitors that differentiate into NK cells with T-bet (like ILC1) and Eomes (eomesodermin; required for NK cell development) expression, and (3) LTi progenitors that differentiate into LTi cells with RORγt expression (like ILC3) [94].

Kobayashi et al. revealed that the epidermis and dermis contain mixed phenotypes of ILC2s and ILC3s (GATA3- TSLPR+ IL-13+ RORγT+ ILCs in epidermis; GATA3 low IL-5- RORγT+ ILCs and GATA3+ IL-5+ IL-13- RORγT- ILCs in dermis), whereas the fatty layer contains typical GATA3+ ILC2s [96]. More importantly, they showed that ILC subsets depend on hair follicle-derived factors for persistence (IL-7 and TSLP) and localization (CCL20–CCR6) and control the skin microbiome by restricting sebaceous gland function via suppression of the Notch signaling pathway involving TNF-α and other TNF receptor ligands expressed on ILCs [96].

Accumulating evidence shows that (1) NK cells and ILC1s (activated by IL-7 and IL-15) contribute to murine Th1-biased ACD development via IFN-γ and TNF-α production, (2) CCR6 and CXCR6 are required for NK cell homing to the ACD-like lesion, (3) NK cells negatively regulate ILC2s that regulate type 1 immune responses driven by NK cells and ILC1s, and (4) NK cells and/or ILC1s may possess hapten-specific memory responses not only in the sensitization phase but also in the elicitation phase [11,95,97,98].

Accumulating evidence also shows that ILC2s (activated by IL-33 in the lung and by IL-18 in the skin [59]) contribute to murine Th2-biased ACD development via IL-13 production [97]. Additionally, Rafei-Shamsabadi et al. reported that (1) ILC2s are the most prevalent in the ear of naïve mice, (2) NK cells increase prior to ILC1s (fewer cells), ILC2, and ILC3 in response to hapten, and (3) ILC subsets produce their respective cytokines. They also reported that (4) mice lacking all ILC subsets show enhanced ear swelling in response to hapten, possibly due to increased numbers of Th1 cells, and (5) mice lacking ILC2s also show an increased response, suggesting a regulatory role of ILC2s in this murine ACD model [99].

As described above, functional analyses of ILCs are a hot topic in understanding ACD pathogenesis [11,95,97,98]. Further studies are necessary to elucidate (1) the existence of an innate memory response (hapten-specific) in the draining lymph nodes (ILC1s and/or NK cells) in patients with ACD and (2) the involvement of ILCs in the iSALT cluster in ACD considering similarities in homing lesions and functions between ILCs and effector T cells.

#### 2.2.7. Roles of Neutrophils in ACD

Neutrophils are among the first immune cells to infiltrate the inflammatory skin lesions in response to C-X-C chemokines including CXCL8 (IL-8), CXCL2, and CXCL10 [100,101]. Using murine models including DNFB and calcipotriol (that stimulate TSLP production to skew Th2), Walsh et al. reported that (1) neutrophils are the first to migrate to the lesion in response to CXCL1 (secreted by damaged keratinocytes via PAR2) via CXCR2, (2) neutrophil depletion attenuates pruritus-evoked scratching, (3) neutrophils are required for skin hyperinnervation, enhanced expression of pruritus signaling, and upregulation of inflammatory cytokines, (4) neutrophils are required for induction of CXCL10 (a ligand for CXCR3; possibly produced by keratinocytes) that promotes pruritus via activation of sensory neurons, and (5) CXCR3 blockade attenuates pruritus [101]. Funch et al. also reported that (1) CD8+ resident memory T cells enhance CXCL1 and CXCL2 induction in the skin and initiate neutrophil infiltration in the epidermis within 12 h after re-exposure to the allergen using multiple mouse models and (2) CXCR1 and CXCR2 antagonism attenuates neutrophil recruitment and ACD flareup [102]. 

Mast cells also contribute to the rapid neutrophil infiltration in response to haptens/allergens by enhancing neutrophil extravasation from the bloodstream to the skin lesion via degranulation of TNF-α into the blood vessels (Figure 5) [103]. Dudeck et al. reported that (1) mast cells exhibit intraluminal protrusion at steady state, (2) mast cell degranulation of TNF-α into the bloodstream occurs in response to haptens/allergens (DNFB, IgE crosslinking, and LPS), followed by priming of circulating neutrophils expressing TNF receptors (TNFR1), (3) primed neutrophils expressing CD11b extravasate via intraluminal crawling, and (4) the intravenous administration of mast cell granules promotes neutrophil extravasation (Figure 5) [103].

Neutrophil infiltration may be blocked by regulatory T cells in early stages of the elicitation phase. Ring et al. reported that (1) neutrophils are dominant in the skin (where the allergen is re-exposed) with few T cells, (2) injection of regulatory T cells 6 h after allergen re-exposure blocks neutrophil infiltration and prevents the subsequent infiltration of dDC and T cells, and (3) regulatory T cells tighten endothelial junctions by inducing intracellular cAMP, followed by blocking inflammatory cell infiltration from the blood [104]. Nrf2 negatively regulates recruitment and accumulation of neutrophils in murine ACD as described in Section 2.1.2. Nrf2 enhances the expression levels of CD36 in macrophages that phagocytose damaged neutrophils, suggesting that M2bs (regulatory macrophages) play a role in downregulating neutrophil accumulation in ACD at the end stages of the elicitation phase [105].

Several groups also reported key roles of neutrophils in ACD development. IL-36 receptor antagonist deficient mice show enhanced contact hypersensitivity via the upregulation of the NFκB and MAPK (mitogen-activated protein kinase) signaling pathways, followed by the secretion of proinflammatory cytokines and the recruitment of neutrophils, dDCs, and T cells. Using this mouse model, Hasegawa et al. reported that neutrophil extracellular trap (NET) formation plays a key role in murine ACD development by showing an attenuated response with a pan-peptidyl arginine deiminase inhibitor (peptidyl arginine deiminase 4, an enzyme required for NET formation) [106]. Neutrophils secrete myeloperoxidase that is associated with cellular damage via oxidation. Strzepa et al. reported that neutrophil-derived myeloperoxidase not only promotes IL-1β production and dDCs activation in the sensitization phase but also drives vascular permeability and enhances inflammatory cell infiltration to the skin lesion in the elicitation phase [107].

The administration of 17,18-epoxyeicosatetraenoic acid, an antiallergic and anti-inflammatory lipid metabolite of eicosapentaenoic acid, attenuates murine and monkey ACD responses to DNFB by inhibiting neutrophil migration via activation of the G protein-coupled receptor 40 (free fatty acid receptor 1) [108]. Targeting neutrophils in ACD and associated pruritus may be meaningful considering recent advances, as described in this section.

#### 2.2.8. Roles of Eosinophils in ACD

Eosinophils are derived from granulocyte-monocyte progenitors that differentiate from hematopoietic stem cells in the bone marrow. Eosinophil progenitor differentiation (from granulocyte-monocyte progenitors) requires C/EBPα, C/EBPε, IRF8, PU.1 (E26 transformation-specific family transcription factor), and GATA-1, suggesting the commonalities with cDC2s, ILC2s, and M2a macrophages, as described in each section. Eosinophil differentiation requires XBP1 (that promotes survival during maturation process) and C/EBPε (at peak), followed by ID2 (inhibitor of DNA-binding 2; that enhances maturation efficiency) [109]. IL-5 and the other 2 β common chain cytokines, IL-3 and GM-CSF, are key for eosinophil expansion and maturation [110]. Eosinophils accumulate at sites of tissue damaged by allergen exposure.

Eosinophils are observed frequently in the gastrointestinal tract. Gurtner et al. reported that (1) an active population of intestinal eosinophils shows a different pattern of transcriptome, surface proteome, and spatial location, as compared with a stable population based on a single-cell transcriptomic profiling of eosinophils and (2) active eosinophils possess antibacterial functions (IFN-γ signaling and MCH-I-restricted antigen processing and presentation) and immune regulatory functions (the expression of co-stimulatory molecules, CD80 and PD-L1, and Th17 suppression) based on a genome-wide CRISPR inhibition screen and functional assays. They also reported that (3) IL-33 maturates active eosinophils in vitro and in vivo, (4) IL-33 promotes active eosinophil accumulation in a mouse colitis model and IFN-γ enhances the effect of IL-33, and finally (5) active eosinophils colocalize with CD4+ T cells based on in situ RNA imaging in colon sections from patients with ulcerative colitis (N = 3), suggesting the heterogeneity of eosinophils in the intestine and the key roles of active eosinophils in colitis [111]. Similar findings may be found in ACD and/or other inflammatory skin diseases.

Eosinophil cationic protein (ECP) is released by eosinophils and is a marker for allergic inflammation including asthma and atopic dermatitis. Kim et al. reported that (1) ACD patients who were confirmed with positive patch testing (N = 169) show a significantly higher value of serum ECP than non-ACD patients (N = 47) with negative or irrelevant results (20.6 (12.9–31.3) μg/L vs. 13.3 (6.7–18.3) μg/L; median (25% percentile–75% percentile)) and (2) demographics (sex; occupation; atopic dermatitis comorbidity; location of eczema; age; disease duration; body surface area involvement) and other biomarkers (eosinophil count; IgE) show no significant differences. They also reported that (3) a multivariate analysis adjusted for age, sex, and serum allergy markers shows a correlation between ECP values and sensitization risk to formaldehyde as an allergen (odds ratio; 1.48; confidence interval 1.11–1.96) [112]. These findings may suggest that eosinophils are involved in ACD pathogenesis via ECP, thereby recruiting other players to the lesion. Formaldehyde is involved in airborne contact dermatitis that may share a similar pathogenesis with asthma.

#### 2.2.9. Roles of Basophils in ACD

Basophils are derived from basophil progenitors that differentiate from hematopoietic stem cells in the bone marrow. Ontologically, there may be common basophil/mast cell precursors or common basophil/eosinophil precursors [113]. Basophils are blood granulocytes that share morphological and functional similarities with tissue-resident mast cells (FcεRI; basophilic granules; histamine release) [114]. Murine basophils can be generated from bone marrow cells with IL-3 or TSLP in vitro. IL-3-elicited basophils possess FcεRI and IL-3 receptor expression and release histamines, leukotrienes, and proteases, whereas TSLP-elicited basophils possess the IL-3 receptor, TSLP receptor, IL-18 receptor, and ST2 (IL-33 receptor) and release IL-4 and IL-6, suggesting basophils are a heterogenous population [113]. Of note, basophils are observed in mice deficient in both IL-3 and TSLP, suggesting that the other key factor(s) need to be elucidated [113]. Basophils play roles in Th2-biased ACD and are involved in pruritus via mediators including histamine, leukotrienes, cytokines, and chemokines (IL-4, IL-13, IL-31, TSLP, platelet-activating factor), proteases (cathepsin S), prostaglandins (PGE2 and PGD2), and neuropeptides (substance P) [114].

Basophils are elevated in skin obtained from atopic dermatitis patients, as compared with healthy volunteers [115]. Basophil infiltration is observed in a calcipotriol mouse model (described in Section 2.2.7) in response to TSLP production in the skin and basophil depletion attenuates skin inflammation, suggesting that basophil-derived IL-4 plays a role in Th2-biased ACD [113]. Basophil infiltration is also observed in patients with chronic spontaneous urticaria and in an IgE-mediated chronic allergic inflammation mouse model and basophil depletion completely blocks inflammation development and attenuates pruritus, suggesting roles for basophils in IgE-mediated skin diseases and pruritus [113].

Hachem et al. reported that (1) basophils do not extravasate from blood to the skin in response to FITC in IL-3 deficient mice and (2) IL-3 produced by T cells activates ALDH1A2 (retinaldehyde dehydrogenase 1 family member A2; an enzyme for retinoic acid production) that is expressed by basophils and enhances retinoic acid production, followed by the subsequent induction of integrins (ITGAM, ITGB2, ITGA2B, ITGB7) that are required for basophil extravasation in the ACD lesion using mice lacking IL-3 selectively in T cells [116]. As described in Section 2.2.5, basophils express FcεRI that bind free IgE but not IgE–immune complex, indicating that basophils are sensitized by free IgE but not IgE–immune complex (using FcεRII/CD23) in response to haptens/allergen [81].

## 3. Discussion

In this review, we discussed ACD and the role of innate immune cell types in ACD pathogenesis. We also discussed the importance of crosstalks between innate immune cells, iSALT, and interactions with T cells or B cells.

Although a murine contact hypersensitivity model is useful to elucidate some aspects of ACD pathogenesis, the murine model is different from ACD in humans due to the following: (1) functions and population densities of specific immune cell types (Th22 cells; γδT cells) are different; (2) surface markers for each immune cell type are not always identical (CD1a for human LCs); and (3) weak sensitizers (nickel) which are usually sufficient to cause ACD in patients cannot induce reactions in mice. These differences indicate the reasons for validation studies using human samples (skin specimens and immune cells including Mo-DCs from blood) and multidisciplinary approaches (e.g., a patch test from healthy volunteers or ACD-susceptible patients) to further elucidate the pathogenesis of ACD [21].

During the development of topical medications, it is of critical importance to identify the potential risk of ACD prior to first in-human clinical trials. If there is potential risk of skin irritation including ACD that has been identified in an animal sensitization study and/or dermal toxicology study, the program may need to be terminated or extensively revised. When animals are used to predict and rule out the risk of ACD in patients during the development of topical drugs (not only for antibiotics but also treatments for skin diseases (acne vulgaris; atopic dermatitis; psoriasis) and systemic diseases), various factors may need to be prespecified: (1) how to determine the dose and dosing regimen and (2) how to mimic human ACD status by optimizing the animal models (genetically or pretreatment with cytokines). Considering the principles of the 3 Rs (replacement; reduction; refinement) for experimental animal research, in vitro assay development using human Mo-DCs and/or serum/plasma may be beneficial to predict ACD-inducing haptens/allergens in ACD-susceptible patients and to study ACD pathogenesis. Additional in vitro or ex vivo studies using human samples prior to clinical trials may be informative to investigate the potential risk of ACD in the development of topical drugs.

In this review, we also discussed the effects of innate immune cells on tumor cells since immuno-oncology drugs may exacerbate ACD possibly via the activation of CD8+ and/or CD4+ T cells and the inactivation of regulatory T cells and tolerogenic DCs. Of note, recent findings associated with cDC2s are intriguing and may be helpful to elucidate the ACD pathogenesis. 

## 4. Conclusions

Innate immune cells include Langerhans cells, dermal dendritic cells, monocytes, macrophages, mast cells, innate lymphoid cells (group 1 (natural killer cells; ILC1); group 2 (ILC2); group 3 (lymphoid tissue-inducer cells; ILC3)), neutrophils, eosinophils, and basophils. These cells play more active roles in ACD than previously thought by using PRRs including TLRs and NLRs. Innate immune cells recognize DAMPs and cascade the signal to produce various cytokines and chemokines including TNF-α, IFN-α, IFN-γ, IL-1β, IL-4, IL-6, IL-12, IL-13, IL-17, IL-18, and IL-23. We discussed the recent findings focusing on the roles of the innate immune system in ACD during the sensitization phase and elicitation phase.

## 5. Future Directions

As discussed above, ACD pathogenesis in humans (or the degree of innate immune involvement in ACD development) is heterogenous, especially depending on haptens/allergens (including concentrations), location of the lesion, baseline demographics (including age, sex, and ethnicity) and comorbidities (including atopic dermatitis), and exposome (environmental factors including nutrition, psychological/lifestyle factors, medication, occupational factors, pollutants, and climate factors). All these factors may need to be well taken into account when both non-human species and human samples are used to elucidate human ACD pathogenesis. Also, when specific immune cell types are investigated, both ontological and functional aspects may need to be considered. Blood-derived cell types like Mo-DCs are more easily accessible than tissue-derived cells, especially in humans.

ACD on the palms of the hands and soles of the feet (palmoplantar area) may cause severe and prolonged itch and pain despite the low incidence, as compared with ACD that is developed on other sites of the body. Little is known about the distribution pattern and functions of skin-resident immune cells of the palmoplantar area, as compared with other body sites; it is well expected that the features of these resident immune cells are unique due to the following differences between the two skin types: (1) the thickness of the epidermis with specific cytokeratin filaments, (2) the density of melanocytes that may have a crosstalk with LCs, (3) the glabrous vs. hair-bearing, (4) the density of eccrine sweat glands, (5) the distribution pattern of peripheral nerves/nerve endings, (6) the different fibroblast gene signatures (associated with Wnt signaling), and (7) the structural difference in size of adipose tissues, possibly all of which are associated with consistent mechanical stress on palms of the hands and soles of the feet. 

Although various intriguing findings are reported focusing on the effects of atopic dermatitis on ACD as discussed above, additional studies are needed to elucidate how to treat ACD in atopic dermatitis and how to perform vaccinations in atopic dermatitis patients in a mechanistically reasonable way.

Gender differences are well recognized in autoimmune diseases. The incidence of ACD is higher in females than males possibly due to daily activities including the frequent use of fragrance and metal [2]. Sex hormones including estrogen and progesterone promote Th2 immunity [117], which may indicate a high incidence of Th2-biased ACD in females. XIST is the long non-coding RNA that establishes X chromosome inactivation in female cells in early development [118] and is continuously required in adult human cells to silence a subset of X-linked immune genes including TLR7 (an X-linked receptor that recognizes single-strand RNA-containing immune complexes that are involved in female-biased autoimmunity; associated with lupus) [119]. The genes associated with skin barrier function (filaggrin and claudin [5]) and detoxification enzymes (arylamine N-acetyltransferases 1 and 2 and glutathione-S-transferases [6]) may be regulated by gender-biased immune responses, although further investigations are necessary. Dvornyk et al. recently reported that filaggrin gene polymorphisms including rs12144049 are associated with atopic dermatitis in females but not males in the Caucasian of Central Russia [120], suggesting that similar findings may be observed in ACD patients.

## Figures and Tables

**Figure 1 ijms-24-12975-f001:**
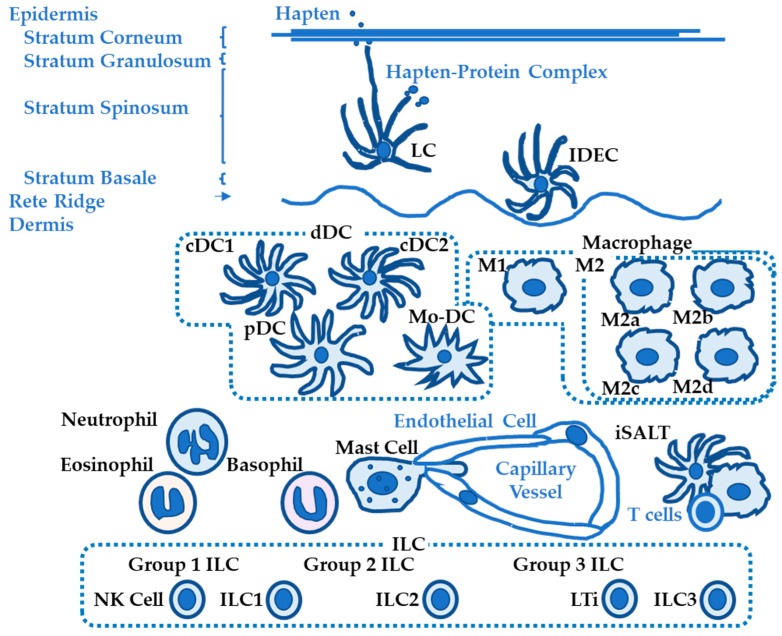
Overview of innate immune cells in allergic contact dermatitis. Key innate immune cells are shown in black. Key substances, structures, and cells are shown in blue. LC = Langerhans cell; IDEC = inflammatory dendritic epidermal cell; dDC = dermal dendritic cell; cDC = conventional dendritic cell; pDC = plasmacytoid dendritic cell; Mo-DC = monocyte derived dendritic cell; M1 = macrophage type 1; M2 = macrophage type 2; iSALT = inducible skin-associated lymphoid tissue; ILC = innate lymphoid cell; NK cell = natural killer cell; LTi = lymphoid tissue inducer cell.

**Figure 2 ijms-24-12975-f002:**
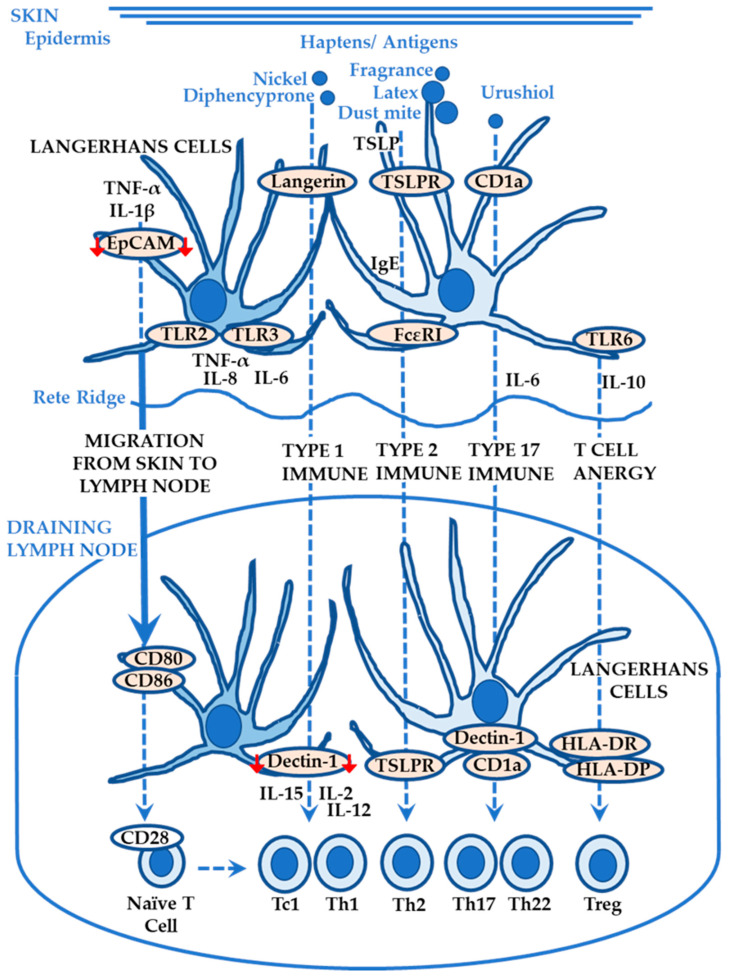
Roles of Langerhans cells (LCs) in allergic contact dermatitis. **Upper** and **lower panels** indicate LCs located in skin and lymph node, respectively. Dotted arrows indicate migration of LCs from skin to lymph node (**far left**), Type 1 immune induction (**2nd left**), Type 2 immune induction (**middle**), Type 17 immune induction (**2nd right**), and T cell anergy pathway (**far right**). Red arrows indicate downregulation. Receptors on LCs are highlighted in orange. TSLP = thymic stromal lymphopoietin; TSLPR = TSLP receptor; CD = cluster of differentiation; TNF = tumor necrosis factor; IL = interleukin; EpCAM = epithelial cell adhesion molecule (CD326); TLR = toll-like receptor; FcεRI = high-affinity IgE receptor; HLA-DR/DP = human leukocyte antigen–DR/DP isotype; Tc1 = type 1 CD8+ T cell; Th = T helper cell; Treg = regulatory T cell.

**Figure 3 ijms-24-12975-f003:**
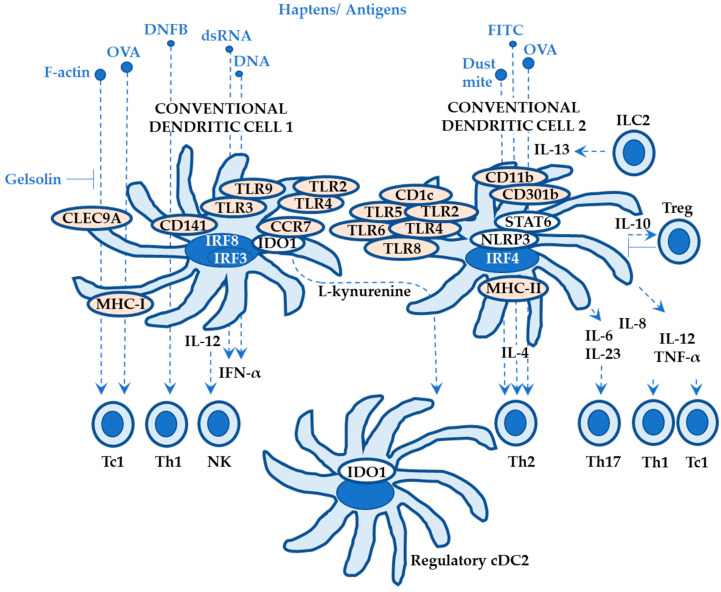
Key roles of conventional dendritic cells 1 and 2 (cDC1s and cDC2s). cDC1s activate Tc1, Th1, and NK cells; cDC2s form a heterogenous population. CCR7+ IDO1+ cDC1s induce regulatory cDC2s via L-kynurenine. Receptors on dermal dendritic cells are highlighted in orange except for NLRP3 located in cytoplasm. Cytoplasmic proteins and transcription factors are highlighted in white and blue, respectively. DNFB = 2,4-dinitrofluorobenzene; FITC = fluorescein isothiocyanate; OVA = ovalbumin; ds = double-strand; CLEC = C-type lectin receptor; TLR = toll like receptor; CD = cluster of differentiation; CCR = C-C chemokine receptor; IDO = indoleamine 2,3-dioxygenase; NLRP = nucleotide-binding oligomerization domain-like receptors family pyrin domain-containing; STAT = signal transducer and activator of transcription; IRF = interferon regulatory factor; MHC = major histocompatibility complex; IL = interleukin; TNF = tumor necrosis factor; IFN = interferon; Tc1 = Type 1 CD8+ T cell; Th = T helper cell; Treg = regulatory T cell; ILC = innate lymphoid cell; NK = natural killer cell.

**Figure 4 ijms-24-12975-f004:**
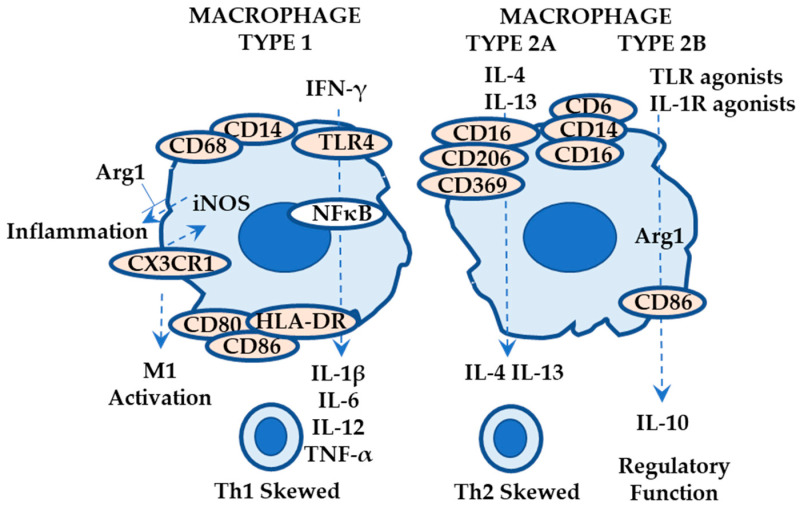
Roles of macrophage type 1 (M1), type 2a (M2a), and type 2b (M2b). M1, M2a, and M2b are associated with Th1, Th2, and regulatory function, respectively. Receptors on macrophages are highlighted in orange. IFN = interferon; IL = interleukin; TNF = tumor necrosis factor; TLR = toll like receptor; CD = cluster of differentiation; CX3CR1 = CX3C motif receptor 1; HLA-DR = human leukocyte antigen–DR isotype; Arg = arginase; iNOS = inducible nitric oxide synthase; NFκB = nuclear factor kappa-light-chain-enhancer of activated B cells; Th = T helper cell.

**Figure 5 ijms-24-12975-f005:**
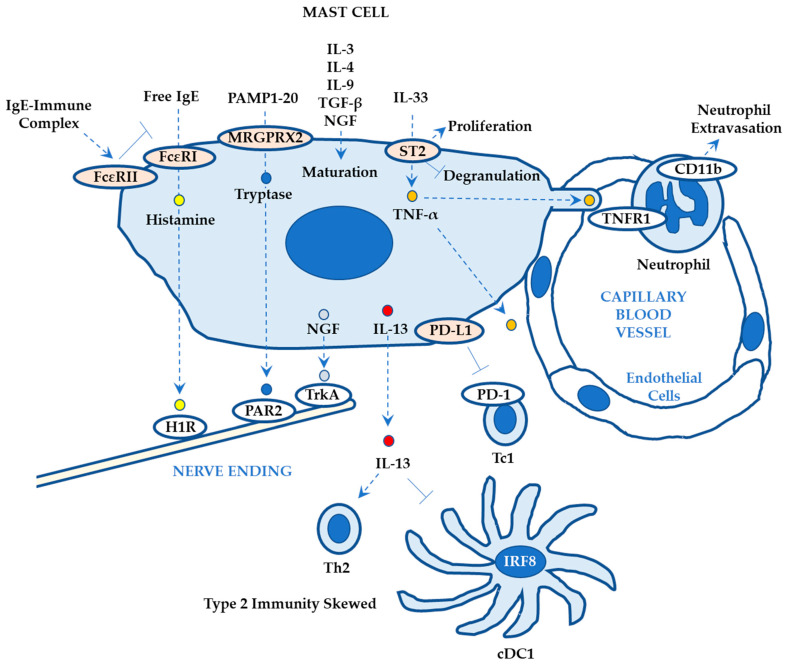
Roles of mast cells. Mast cells play a key role in pruritus via IgE–FcεRI–histamine–H1R axis and PAMP1-20–MRGPRX2–tryptase–PAR2 axis. IL-33 is involved in mast cell growth, suppression of degranulation at chronic phase, and production of TNF-α that is involved in neutrophil extravasation. Mast cells also play a central role in type 2 immunity. Receptors on mast cells are highlighted in orange. FcεRI = high-affinity IgE receptor; FcεRII = low-affinity IgE receptor; MRGPRX2 = Mas-related G-protein-coupled receptor X2 (mouse ortholog Mrgprb2); ST2 = IL-33 receptor; PD-L1 = programmed death ligand 1; PAMP1-20 = proadrenomedullin N-terminal 20 peptide (MRGPRX2 agonist); IL = interleukin; TGF = transforming growth factor; NGF = nerve growth factor; TNF = tumor necrosis factor; CD = cluster of differentiation; H1R = histamine receptor 1; PAR2 = protease-activated receptor 2; TrkA = tropomyosin receptor kinase A; PD-1 = programmed death-1; IRF8 = interferon regulatory factor 8; yellow circles (**far left**) = histamine; blue circles (**2nd left**) = tryptase; gray circles (**middle**) = NGF; red circles (**2nd right**) = IL-13; orange circles (**far right**) = TNF-α; cDC1 = conventional dendritic cell 1; Tc1 = type 1 CD8+ T cell.

**Figure 6 ijms-24-12975-f006:**
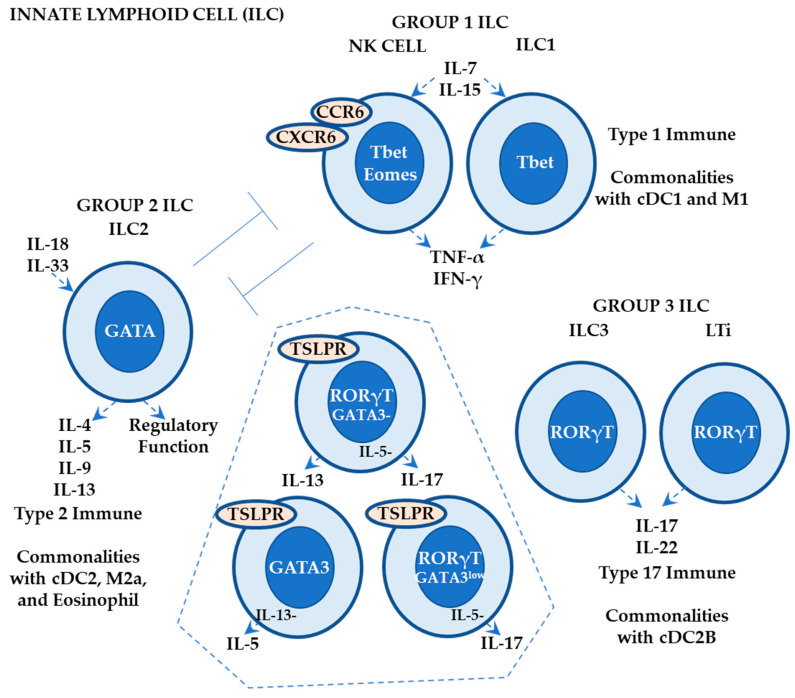
Roles of Innate Lymphoid Cells (ILCs). Groups 1, 2, and 3 ILCs are associated with Type 1, Type 2, and Type 17 immune, respectively. NK cells suppress ILC2s function. ILC2s suppress Type 1 immune response. Dotted areas indicate ILCs with mixed phenotypes of ILC2 and ILC3 exist in epidermis and dermis. Receptors on ILCs are highlighted in orange. CCR = C-C chemokine receptor; CXCR = C-X-C chemokine receptor; TSLPR = thymic stromal lymphopoietin receptor; IL = interleukin; TNF = tumor necrosis factor; IFN = interferon; T-bet = T-box expressed in T cells; Eomes = eomesodermin; ROR = retinoid orphan receptor; cDC = conventional dendritic cell; M1/2a = macrophage type 1/2a; LTi = lymphoid tissue inducer cell.

## Data Availability

Not applicable.

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
