# Peer review of "Role of Innate Immunity in Allergic Contact Dermatitis: An Update"

_ijms, 2023, doi:10.3390/ijms241612975_

Round 1
Reviewer 1 Report
The article is interesting but a little difficult to follow
Author Response
The article is interesting but a little difficult to follow.
REPLY: We thank the reviewer for this constructive comment. We have revised all the figures (Figures 1 – 5) and added new Figure 6. We hope these figures with legends describing brief summaries would be helpful for the reviewers/ readers to follow the text as much as possible.
Reviewer 2 Report
The review article entitled “A Role of Innate Immunity in Allergic Contact Dermatitis: an Update” is informative.
Comments:
In the light of a higher incidence of allergic contact dermatitis in females, it is reasonable to suppose the involvement of X-linked immune response.
As was pointed out in section 5, is there any supporting evidence suggesting that the genes associated with skin barrier function such as filaggrin are regulated by gender-biased immune responses?
Author Response
In the light of a higher incidence of allergic contact dermatitis in females, it is reasonable to suppose the involvement of X-linked immune response. As was pointed out in section 5, is there any supporting evidence suggesting that the genes associated with skin barrier function such as filaggrin are regulated by gender-biased immune responses?
REPLY: We thank the reviewer for this constructive and intriguing comment and added the Science paper published in 1999 (Lns 964 – 967 with no markup version; Lns ;972 – 975 with all markup version with track changes; Reference 117 as highlighted in red; indicating gender differences in autoimmune diseases) and the PLoS One paper published in 2021 (Lns 975 - 978 with no markup version; Lns ;983 - 986 with all markup version with track changes; Reference 120 as highlighted in red; showing the SNPs of filaggrin genes in women but not in men in atopic dermatitis patients from Central Russia Caucasian).
Reviewer 3 Report
This is a well-written manuscript. The comments are attached in pdf.

Author Response
Line 11: This paper should be included: Annunziato F, Romagnani C, Romagnani S. J Allergy Clln lmmunol.2015 Mar;135{3):626-35.
REPLY: We thank the reviewer for this constructive comment. We included the JACI paper (Reference 93 as highlighted in red) and revised the abstract (Ln 12; Ln 13; Ln 21; Ln22) and the main text (Lns 707 – 711 with no markup version; Lns 784 – 788 with all markup version with track changes) as highlighted in red accordingly.
In the introduction, the authors list what are the risk factors for CD and ACD, maybe it would be worth enumerating here the occurrence of metal hypersensitivity and implanted devices (knee, dental etc) PMID: 36498546, and not only skin exposures.
REPLY: We thank the reviewer for this constructive comment. We included the J Clin Med paper (Reference 4 as highlighted in red) in the Introduction section (Lns 31 – 33 with or without track changes) as highlighted in red.
The figures are very hard to understand. I suggest working on those. The manuscript would benefit. There is much text, and good figures would be helpful. e.g. Fig2. All abbreviations used on the figure should be explained, Here we miss a lot: LC, IL, Th, TNF....
REPLY: We thank the reviewer for this constructive comment. We have revised all Figures (Figures 1 – 5) accordingly: we have added thorough abbreviations/ acronyms throughout the Figures. We also explained the schemes as much as possible as highlighted in red with track changes.
As for Figure 2, wavy line indicates “rete ridge” and revised Figure 1 and Figure 2. The line of lymph node was fully inserted in the revised Figure 2. Each arrow was explained in the revised Figure 2.
Also, we have generated Figure 6 (ILCs) to explain the text with thorough abbreviations/ acronyms with detailed explanations.